# Insect-habitat-plant interaction networks provide guidelines to mitigate the risk of transmission of *Xylella fastidiosa* to grapevine in Southern France

Xavier Mesmin[1,☉]*, Marguerite Chartois[1,☉], Pauline Farigoule[1], Christian Burban[2], Jean-Claude Streito[1], Jean-Marc Thuillier[1], Éric Pierre[1], Maxime Lambert[1], Yannick Mellerin[2], Olivier Bonnard[2], Inge van Halder[2], Guillaume Fried[3], Jean-Yves Rasplus[1], Astrid Cruaud[1‡], Jean-Pierre Rossi[1‡]

**1** CBGP, INRAE, CIRAD, IRD, Institut Agro, Univ Montpellier, Montferrier-sur-Lez, France, **2** BIOGECO, INRAE, University of Bordeaux, Cestas, France, **3** Anses, Laboratoire de la Santé des Végétaux, Unité Entomologie et Botanique, Montferrier-sur-Lez, France

☉ These authors contributed equally to this work.
‡ AC and JPR also contributed equally to this work.
* xavier.mesmin@mailo.fr

## Abstract

*Xylella fastidiosa* (*Xf*) is a xylem-limited bacterium that has been recorded in several European countries since its detection in 2013 in Apulia (Italy). Given the prominence of the wine industry in many southern European countries, a big threat is the development of Pierce's disease in grapevines. Yet, the insect-habitat and insect-plant interaction networks in which xylem feeders, possible vectors of *Xf*, are involved around European vineyards are largely unknown. Here we describe these networks in three key wine-growing regions of southern France (Provence-Alpes-Côte d'Azur, Occitanie, Nouvelle-Aquitaine) to identify primary xylem feeder habitats, and assess their specialization degree at the habitat, plant family, and plant species levels. A total of 92 landscapes (and 700 sites) were studied over three sampling sessions in the fall 2020, spring 2021, and fall 2021. Among the habitats sampled, meadows hosted the largest xylem feeder communities, followed by alfalfa fields. Vineyard headlands and inter-rows hosted slightly smaller xylem feeder communities, indicating that potential *Xf* vectors thrive near crops. Grapevines hosted few xylem feeders, suggesting rare but possible transfer to vulnerable crops. *Philaenus spumarius*, *Aphrophora alni*, *Lepyronia coleoptrata*, and *Cicadella viridis* were all similarly generalists at the habitat, plant family or plant species level. The only specialist was *Aphrophora* grp. *salicina*, which was restricted to riparian forests and to Salicaceae. *Neophilaenus* spp. were extremely specialist at the plant family level (Poaceae), but rather generalist at the habitat and plant species levels. All 1017 insects screened for the presence of *Xf* tested negative, showing that *Xf* is not widespread in the studied regions. Our study provides new basic ecological information on potential vectors of *Xf*, especially

**Data availability statement:** Data and scripts are freely available on the public Zenodo repository at: https://doi.org/10.5281/zenodo.11181989.

**Funding:** All authors were funded by Hennessy (https://www.hennessy.com). The funder enabled access to the vineyards of adherent winegrowers. Besides, the funder was not involved in the study design, data collection and analysis, decision to publish, or preparation of the manuscript.

**Competing interests:** The authors have declared that no competing interests exist.

on their specialization and feeding preferences, as well as practical information that may be relevant for the design of epidemiological surveillance plans.

## Introduction

*Xylella fastidiosa* (*Xf*) (Xanthomonadaceae) is a xylem-limited bacterium transmitted by hemipteran insect vectors (mainly sharpshooters and spittlebugs) [1,2] that causes various diseases in a broad range of cultivated and uncultivated plants [3,4]. *Xf* originates from the Americas but has been introduced to different regions of the world during the last decades (https://gd.eppo.int/taxon/XYLEFA/distribution).

Genetic studies revealed that *Xf* is divided into five subspecies [5–7], each one with a slightly different host range. *Xf* subsp. *sandyi* is responsible of oleander leaf scorch, while *Xf* subsp. *multiplex* develops on a large range of host plants, from alfalfa to almond tree. *Xf* subsp. *pauca* develops on various host plants, but its main agronomic impact is on Citrus and Olive industries. The subspecies *pauca* causes more than 100 million USD worth of losses each year to the Brazilian citrus industry through the Citrus Variegated Chlorosis [8], and a loss over 50 years that could reach 5.2 billion EUR (5.6 billions USD) on the European olive industry through the olive quick decline syndrome [9]. *Xf* subsp. *sandyi* is supposed to be a recombinant between subsp. *fastidiosa* and subsp. *multiplex*. And finally, *Xf* subsp. *fastidiosa* is the causal agent of Pierce's disease in grapevines, which causes millions of losses to the US grape industry every year [10]. In France, the wine and spirits sector is prominent, with annual export profits estimated to exceed 10 billion euros [11]. The epidemic risk posed by *Xf* is therefore a legitimate concern for the french wine industry, especially since the subspecies *fastidiosa* has recently been detected in southern Italy [12–14].

The insect vectors of *Xf* are xylem feeders, and belongs to sharpshooters and spittlebugs species. In the US, the main vectors are *Graphocephala atropunctata* and *Homalodisca vitripennis* [15,16]. In Europe, the proven vectors are *Philaenus spumarius* [17–19], *P. italosignus* and *Neophilaenus campestris* [20]. Other candidates vectors are *N. lineatus*, *Cicadella viridis*, *Aphrophora* spp. and *Lepyronia coleoptrata* [16,21], also common European species of xylem feeders, frequently observed in surveys conducted in agrosystems [22–24].

When an insect feeds on an infected plant and acquires the bacterium, it can transmit it to the next plant it feeds on, without a latency period [25]. The probability of contaminating a healthy crop is thus linked to the connectance of the trophic network composed of the crop, the vectors, and the contaminated plants surrounding the crop. This connectance somewhat reflects vector diet breath in different types of habitat. Therefore, to better assess the risk of contamination by *Xf*, it is essential to decipher habitat-plant-vector interactions at the landscape level around crops.

We still have little knowledge on these networks of interaction. In the US, riparian forests were proven favorable to *Graphocephala atropunctata*, which is responsible for most of the primary transmission of *Xf* to neighboring vineyards [26]. As a consequence, greater damage due to Pierce's disease was observed near rivers [27]. In Belgium, Casarin et al. [28] suggested that Salicaceae, abundant on river banks,

hosted populations of vectors of *Xf* (especially *Aphrophora* grp. *salicina*). Salicaceae could therefore be a key component of networks through which *Xf* could spread. In France, reports from Corsica also indicated large populations of *Aphrophora* grp. *salicina* and *Cicadella viridis* in riparian forests [21]. *Philaenus spumarius*, which is considered as the main *Xf* vector in Europe [29], is particularly abundant in typical scrubland of the Mediterranean area. It is often found in strong association with *Cistus monspeliensis* [30] or *Dittrichia viscosa* [24]. Large populations of *P. spumarius* were also reported in grasslands [31] and alfalfa fields [32]. Alfalfa and red clover are indeed the only crops for which *P. spumarius* is considered a pest through direct feeding on plant sap (e.g., [33]). But except in Corsica where *P. spumarius* shows a strong preference for a few plant species, it acts as a super-generalist species in other territories and can be found in various habitats (herbaceous, olive groves, and shrubs of orchards borders) [31,34] and numerous plant species and botanical families [35]. Species in the genus *Neophilaenus* are known to specialize in Poaceae [23,35] and are very abundant in herbaceous habitats, especially in grasslands [31] but also in the herbaceous cover and headlands of groves [24,34]. So far, european studies on *Xf* vectors have mainly been conducted in natural landscapes or orchards, with the exception of the study by Cappellari et al. [31] conducted in a Mediterranean landscape including vineyards. Thus, almost nothing is known about the habitat-plant-vector networks in temperate wine-growing areas of Europe, which prevents any investigation on which levers we could act to prevent or mitigate transmission risk of *Xf* to grapevine.

In mainland France, *Xf* has been detected by official surveillance of plants in two wine-producing regions in southern France: Provence-Alpes-Côte d'Azur (PACA) and Occitanie (OCC). The subspecies *multiplex* is by far the most commonly detected. Only one focus of the subspecies *pauca* has been identified so far, near Menton [36] and is now eradicated [37]. *Xf* subsp. *fastiodiosa* has never been detected. *Xylella* (unidentified subspecies) has been fortuitously found once in vine bark in the Lussac Saint-Émilion wine-growing region [38]. Although this result needs to be confirmed, it suggests that the bacterium might be present at very low prevalence levels in a third wine-producing region of France, Nouvelle-Aquitaine (NAQ), for several years.

In the present study, we characterized insect-habitat-plant interaction networks in the three above mentioned wine-growing regions of France. We addressed the following questions: 1/ What habitats are the biggest reservoirs of xylem feeders? and 2/ How specialized are xylem feeders at the habitat and host plant levels? Based on the literature, we expected that 1/ the biggest reservoirs would be mainly semi-natural herbaceous strata (meadow, alfalfa, riparian vegetation) [28,32]; 2/ *P. spumarius* would be the most generalist at the habitat and host plant levels whereas other xylem feeders would be mostly specialists such as *Neophilaenus* spp. in meadows, or *Cicadella viridis* and *Aphrophora* spp. in riparian habitats [28,39,40].

In addition, we assessed whether or not *Xf* was present in the studied environment by relying on a sentinel insect approach. It relies on the idea that as vectors feed on various plants, their infectious status somewhat summarizes the infection status of the environment where they have been sampled [14,41–43]. This strategy is an interesting addition to surveillance based on symptomatic plants, because it can reveal overlooked contamination [41,42]. Indeed, while in certain species the proliferation of *Xf* can lead to plant death, in many species, the bacterium is commensal, and plants do not develop symptoms [6,44]. In this case, asymptomatic habitats can act as disease reservoir posing a threat to nearby susceptible crops. Given the results obtained by official surveillance we would expect that *Xf* could spread within insect-habitat-plant interaction networks in the Mediterranean region. However, if present, the prevalence of *Xf* in vector populations of the NAQ region would be low.

## Methods

### Study sites and sampling design

Our sampling strategy focused on a set of vineyards in three French wine-producing regions: Nouvelle-Aquitaine (NAQ), Occitanie (OCC), Provence-Alpes-Côte d'Azur (PACA) (Fig 1). Since no regulation applies to proven or putative insect vectors of *Xf* in France, and since none of our sampling sites fell within any protected area, no permit delivered by French

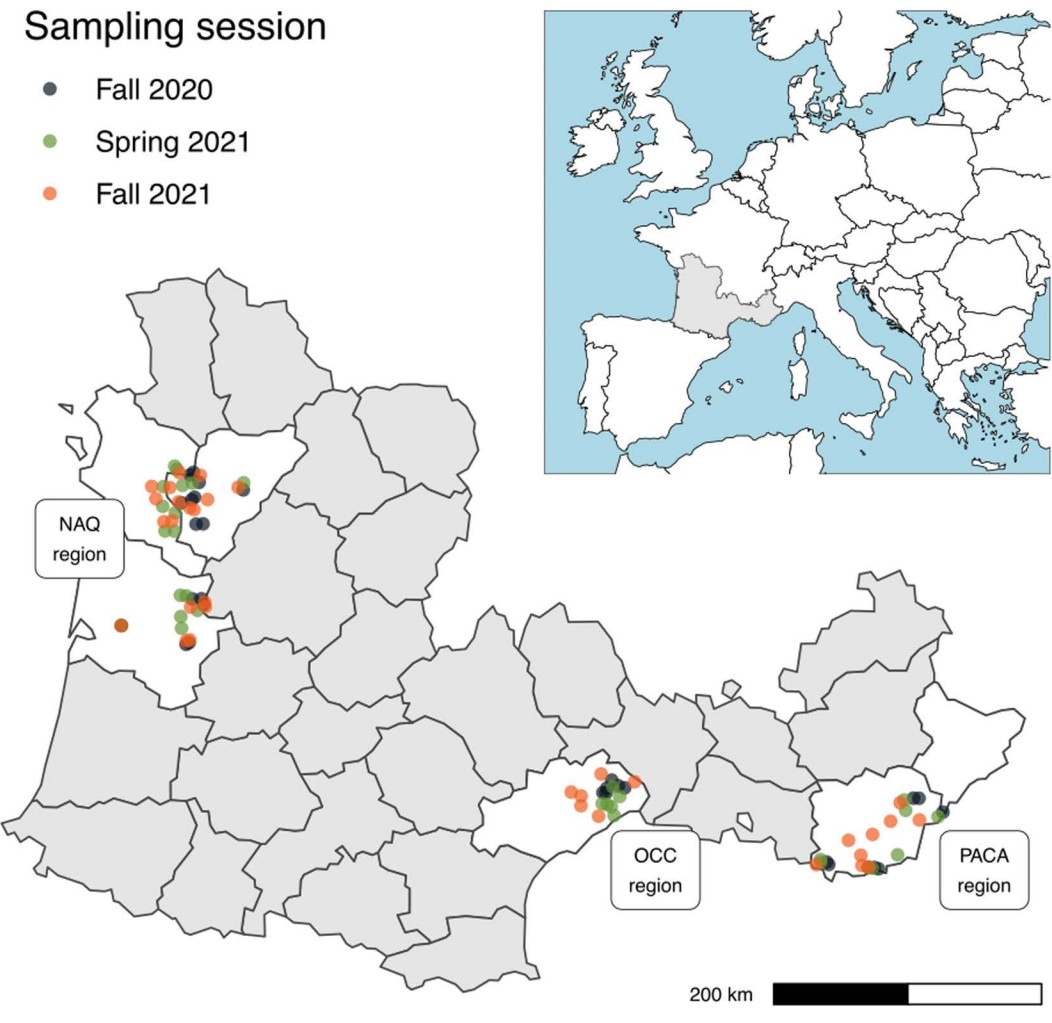

**Fig 1. Map of the studied areas.**

authorities was necessary to conduct the sampling. For samples made in farmland, farmers were contacted before field visits through the network of our funder, and all granted access to their fields.

The grey area on the inset shows the sampled area in France, and the zoomed-in main figure highlights in white the areas where buffers (colored dots) were placed. Country contours were displayed using Natural Earth (version 5.1.1) under a CC0 licence, French departments were displayed using ADMIN EXPRESS database from the National Institute of Geography (IGN), freely available under a CC BY licence.

We used a clustered sampling strategy (Table 1, S1 Appendix) based on the dispersal ability of the main vectors of *Xf*, which is some hundreds of meters in their whole life, and exceptionally more than 1 km [45–47]. We defined 1 km-radius buffers regularly distributed among the three studied areas (Fig 1). These buffers included target vineyards and at least two types of land cover that could be reservoirs of *Xf* vectors according to the literature and our personal observations. These land cover types included olive groves, alfalfa fields, forests, meadows, riparian zones, and the borders of the vine-yards and olive groves sampled. The number and identity of land cover types varied between buffers depending on local landscape composition. Land cover types within each buffer were selected based on their prevalence and accessibility,

**Table 1. Sample size and full description of the clustered sampling strategy used in the survey.**

| Session | Region | Number of buffers | ⊃ | Number of sites | ⊃ | Number of subsites | Number of xylem feeders | Number of insects analysed for the presence of *Xf* |
|---|---|---|---|---|---|---|---|---|
| Fall 2020 | NAQ | 15 | | 111 | | 169 | 1426 | 464 |
| | OCC | 6 | | 49 | | 78 | 83 | – |
| | PACA | 8 | | 64 | | 97 | 796 | 93 |
| Spring 2021 | NAQ | 16 | | 108 | | 161 | 1047 | – |
| | OCC | 6 | | 43 | | 72 | 79 | – |
| | PACA | 8 | | 58 | | 93 | 619 | – |
| Fall 2021 | NAQ | 18 | | 127 | | 191 | 3159 | 401 |
| | OCC | 6 | | 55 | | 86 | 286 | – |
| | PACA | 9 | | 85 | | 132 | 1277 | 59 |
| Total | | 92 | | 700 | | 1079 | 8772 | 1017 |

with no fixed rule regarding the number of patches per land cover type. Inside each buffer, we sampled a median number of 7.5 sites per buffer, each site corresponding to a specific land cover type. Within each site, insect sampling was conducted in distinct vegetation strata: the lower stratum (below 1m, mostly herbaceous vegetation) and, when present, the upper stratum (1 to 2m above ground, mostly woody vegetation). In the context of our study, whe definie a "habitat" as the combination of a vegetation stratum and a land cover type (e.g. "lower stratum of vineyards" or "upper stratum of vineyards").

Buffers comprise four to thirteen sites belonging to the land cover types analyzed. Each site includes two subsites (lower and upper strata, when available) belonging to a particular "habitat". The sampling unit (i.e. on which the sampling protocol was applied) is the subsite. "x ⊃ y" stands for "x includes y" and denotes the clustered sampling strategy with several sites per buffer and two subsites (upper and lower stratum if available) per site. The two last columns show the number of xylem feeders sampled as well as the number used for *Xf* screening.

Three sampling sessions were performed: two targeting the adults of xylem feeders, in the fall of 2020 (14th Sept.-12th Oct.) and 2021 (6th-28th Oct.), and one targeting the nymphs in the spring of 2021 (12th Apr.-9th May). Phenological surveys conducted throughout the year 2021 (S2 Appendix) in OCC and NAQ regions show that nymphs were sampled at their peak densities, and that adults were sampled at their second highest peak of the year. These sampling periods are thus pertinent regarding insect phenology, although a limitation of our study is that adults were not sampled in mid-May, when they reach their highest densities. New buffers were selected for each sampling session. This choice was made to generate as many replications as possible in time and space, capturing most of the variability in insect preferences in the three contrasted regions studied. This was preferred over repeatedly sampling the same buffers and sites (but see S2 Appendix for a different choice regarding the particular case of longitudinal surveys), a design that would have generated pseudoreplication [48] and overall limited our inferential potential at the regional level. The median shortest distance between two buffer contours was 3.3 km, and 90% of the buffers were at least 1.0 km away from the nearest buffer. Within a given sampling session, buffer contours were at least 1.1 km away, with a median shortest distance of 6.1 km, ensuring that we indeed sampled different populations.

### Insect and plant sampling

For the autumnal adults, interaction networks were evaluated at the habitat level only. Adults were collected by passing a sweep net through the vegetation for a total of four minutes. The four minutes were divided into eight periods of 30-second of sweep netting followed by insect collection in the net using a mouth aspirator (the time needed for insect collection was

not included in the 4'). At the end of the four minutes, insects in the mouth aspirator were killed using ethyl acetate, stored in 75° ethanol, and brought to the laboratory for identification under a binocular microscope using the Biedermann and Niedringhaus [49] identification key. They were then stored in 96° ethanol at 4°C. Each subsite (i.e. lower or upper stratum of a particular site) was sampled only once.

For the nymphs, interaction networks were evaluated at the habitat and the host plant levels (family and species). Identification of feeding plants was possible for Aphrophoridae species because the nymphs produce spittle masses when feeding on plants, providing information on host plant interactions. Nymphs were sampled using a quadrat procedure in each subsite. *C. viridis*, which does not produce spittles masses, was not sampled in spring.

To account for differences in vegetation density (dense or sparse) among lower and upper strata, we adapted the size of quadrats to sample a similar amount of vegetation in all strata. This was done by fixing the lower strata quadrats to $1\ m \times 0.25\ m$ [34] and choosing upper quadrat sizes so that the sampling effort in terms of time spent was the same when no spittle was found. This means that the time spent to examine the foliage (overturn leaves/ isolated clumps of grass), not considering the time needed to sample nymphs in spittles, was the same. We found that a good compromise was to increase the size of upper strata quadrats to $2\ m \times 0.5\ m$, and the upper strata of vineyards to $5\ m \times 0.5\ m$ as vegetation cover was usually very low in April-May (S3 Appendix). Three quadrats were positioned randomly in each subsite. Datasets were summed at the subsite level for data analysis.

For each spittle detected in the quadrat, the nymph(s) was/were collected with a paintbrush and placed in vials filled with 75° ethanol. They were later identified in the lab following Stöckmann et al. [50] and Mesmin et al. [24]. *Neophilaenus campestris* and *N. lineatus* were found morphologically indistinguishable at the nymph stage. Therefore, nymphs of this genus were pooled in the *Neophilaenus* sp. group. Similarly, no morphological features are currently reliable to distinguish nymphs of *A. salicina* from those of *A. pectoralis* [51]. All nymphs were thus labelled *Aphrophora* grp. *salicina*. The plant on which the spittle was found was collected and placed in a ziploc bag together with the insect tube, and identified in the lab following Tison & de Foucault [52], Tison et al. [53] and Eggenberg & Möhl [54]. The nomenclature follows the GBIF Backbone Taxonomy [55], checked using the R package 'traitdataform' [56] to detect synonyms.

### *Xf* prevalence in insects

Screening of *Xf* was performed in a subset of the adults collected: 1017 individuals were chosen in the two most contrasted regions regarding *Xf* infection status, i.e. PACA, where *Xf* has been found as early as 2015, and NAQ where *Xf* has never been detected by official surveillance so far (S1 Appendix). We selected insects collected in the vineyards and vineyard borders belonging to the network of growers of our funder or in buffers belonging to INRAE. Areas in which *Xf* screening was performed are shown in S1 Appendix. We covered most of the area around Cognac (NAQ), included samples from semi-natural habitats around Bordeaux (NAQ) and a buffer 25 km away from official detections in PACA [57]. For molecular analyses, screened populations were defined as groups of 30 insects from the same species, sampled in the same habitat of the same buffer. Adults in all populations of xylem feeders sampled in 2020 were screened for *Xf*. All specimens tested negative for *Xf*. Therefore, in 2021, we chose to screen only populations of *P. spumarius* and *N. campestris* which are known vectors of *Xf* [20], as well as *N. lineatus*, whose vector status has not been assessed yet but which is closely related to *N. campestris*.

*Xf* screening was performed using the protocol described in detail in Farigoule et al. [42]. Briefly, DNA was extracted from individual specimens following Cruaud et al. [41] to minimize the impact of PCR inhibitors. *LeuA*, one of the housekeeping genes of *Xf* [58] was amplified using the two-step PCR approach described in Farigoule et al. [42]. Four replicates were performed for each specimen and a unique combination of indexes was used for tagging all PCR products obtained from the same specimen. Sequencing was performed on a MiSeq system ($2 \times 250\ bp$). The complete pipeline to analyze raw sequence data is available from Farigoule et al. [42] [https://github.com/acruaud/prevalenceXfinsectclimate_2022].

## Ecological data analysis

Data analyses were performed using the programming language "R" [59]. Networks and associated "species-level" [60] metrics were computed using the package 'bipartite' [61].

## Construction of three types of interaction networks by spatial resource level

Three types of interaction networks were separately computed, that varied in the definition of the resource level, because we expected that insect specialization could occur at various levels depending on the species: the habitat, the plant family or the plant species. In habitat networks, the strength of the interaction between an insect species and a habitat was the total number of insects of that species found in this habitat for a given region and a given sampling session. The same logic applies to insect-plant family or insect-plant species networks.

## Aggregation of the data

Insect-plant family and plant species networks were based on nymph data and were computed only in the spring of 2021. Insect-habitat networks were built for each sampling season: nymph data in the spring of 2021, and aggregated data of adults in autumn 2020 and autumn 2021.

Networks were initially computed at the buffer level, but they proved too small to allow robust analysis, with too few interactions and many singletons or doubletons (i.e., species represented by only one or two individuals). Therefore, we computed each type of networks at the grain of the region (NAQ, OCC, PACA) to have at least 20 individuals per insect species in most cases. Indeed, comparing species specialization levels is notoriously challenging [62], especially when small networks are analysed [63]. To avoid misinterpretation of specialization levels, we removed singletons and doubletons from each network [64].

We obtained three insect-plant species-level networks (one per region, nymphs only), three insect-plant family-level networks (one per region, nymphs only), and six insect-habitats networks (one per region, for both nymphs and adults).

## Computation of insect specialization metrics

Furthermore, to respect precautions when comparing species specialization levels, we used different specialization metrics. We computed the Resource Range (RR) and the Paired Difference Index (PDI). The RR is based on presence-absence data and estimate the specificity of associations. The PDI uses quantitative data (link weights) and quantifies the intensity of associations specificity. Both metrics range from 0 (maximum generality) to 1 (maximum specificity) and are known to efficiently segregate specialists from generalists including in in low-connectivity networks, which can happen in small networks [65]. We also calculated the normalised Blüthgen $d'$ [66] and the Species Specificity Index (SSI, [67]). The $d'$ is a measure of association specificity, based on the deviation of realized interactions from the number of possible interactions and ranging from 0 (maximum generality) to 1 (maximum specificity). SSI is a coefficient of variation of resource use. Both of these quantitative metrics are less sensitive to differences in abundance among consumers [63]. Finally, in addition to the raw specialization metrics, following Lami et al. [64], we used the z-transformed metrics computed relatively to the distribution of the metric in 1000 random networks drawn from fixed marginal totals. This approach helps to disentangle specialization and sample sizes [68]. Because $d'$ already integrates a confrontation to a null model [68], only the RR, PDI and SSI were analyzed using z-scores. All metrics increase with specialization, either in absolute value (raw scores), or relative to the probable distribution of values given the sampling design (z-scores).

For all specialization metrics involving a null model in insect-habitat networks, marginal totals were set proportional to the sampling intensity of habitats in the corresponding networks (see S3 Appendix for details) using the function 'mgen' of the package 'bipartite' [61]. For the insect-plant family and species networks, marginal totals were only realized interactions (function 'rd2table').

## Statistical analyses

To address our first hypothesis (the biggest reservoirs would be mainly semi-natural herbaceous strata), we modeled xylem feeder abundance (sum of all species) and diversity (, set to NA when no xylem feeder was found at all) in each subsite using GLMMs, with formula (1). The metric used for diversity was the effective number of species, i.e. the exponential of Shannon index [69], because it balances richness and evenness without favoring common or rare species, and is more straightforward to interpret than the raw Shannon index. Indeed, the effective number of species is the virtual number of species if the community sampled was perfectly even [69].

$$Y_{i,j,k} \sim Region_i + Stage_j + Habitat_k + (1|Buffer_{i,j}/Site_{i,j}) \tag{1}$$

where $Y_{i,j,k}$ is the abundance of an insect species metric on region $i$ on session $j$ for habitat $k$. Random effects were added to consider that metrics computed on the same buffer and the same site (upper and lower strata) were not independent. In $(1|Buffer_{i,j}/Site_{i,j})$, the slash "/" indicates that sites are nested within buffers, in accordance with the sampling design. Note that since buffers and sites were changed for each sampling session, the random effects also account for time dependence in the sampling design.

To address our second hypothesis (insect specialization), we analyzed specialization metrics. The seven specialization metrics were considered as repeated measures of the same underlying phenomenon (i.e. specialization), applied to the same insect species within the same networks. Our main analysis therefore integrated all seven metrics to get a synthetic overview of insect specialization, based on complementary indices that vary in their sensitivity to network properties (e.g. connectance, relative abundances). We performed three GLMMs [70] – one for insect-habitat, one for insect-plant families and one for insect-plant species networks – with the following formula:

$$Y_{i,j,k,l} \sim Species_k + Metric_l + (1|ID_{i,j} : Species_k) \tag{2}$$

where $Y_{i,j,k,l}$ is the insect specialization metric $l$ computed on region $i$ on session $j$ for species $k$. Since pooling all specialization metrics within a single model could artificially inflate statistical power, we included a random effect on the interaction between network ID (i.e. the identifier of the region crossed with the sampling session) and insect species. Multiple specialization metrics computed for the same insect species within the same network are thus statistically treated as repeated measures, thereby controlling for potential inflation in statistical power [71, chapter 11]. For all GLMMs performed in this study, we followed the modeling procedure detailed in S5 Appendix. Briefly, models were fitted using the simplest distribution and link yielding a correct model regarding basic modeling assumptions tested with the package 'DHARMa' [72]. Once fitted, they were analyzed using an analysis of deviance tables (package 'car' [73]). All variables were kept, irrespective of significance, as advised by statisticians [74]. At the end of each modeling procedure, pairwise comparisons among the modalities of significant fixed terms were performed using the packages 'emmeans' [75] and 'performance' [76] and marginal $R^2$ (i.e. part of variance explained by fixed effects) were computed [77].

Additionally to network metrics, raw abundances of *P. spumarius*, *Neophilaenus* sp. and *C. viridis* were analyzed with GLMMs to assess their habitat preferences using formula (1).

## Results

Overall, we sampled 8772 xylem feeders belonging to, in decreasing abundance: *P. spumarius*, *N. campestris*, *N. lineatus*, *C. viridis*, *A.* grp. *salicina*, *L. coleoptrata* and *A. alni* (Fig 2, S1 Appendix). Except for 1 individual collected in the PACA region, *N. lineatus* was only encountered in the NAQ region (in similar proportion with *N. campestris*; 669 *N. campestris*/ 680 *N. lineatus*, Fig 2). *Cicadella viridis* and *L. coleoptrata* were absent from samplings conducted in the OCC region. At the nymph stage, *L. coleoptrata* was only collected in the NAQ region (S4 Appendix). For the statistics described below, all details on $\chi^2$, degrees of freedom and p-values are available in the tables of S5 Appendix.

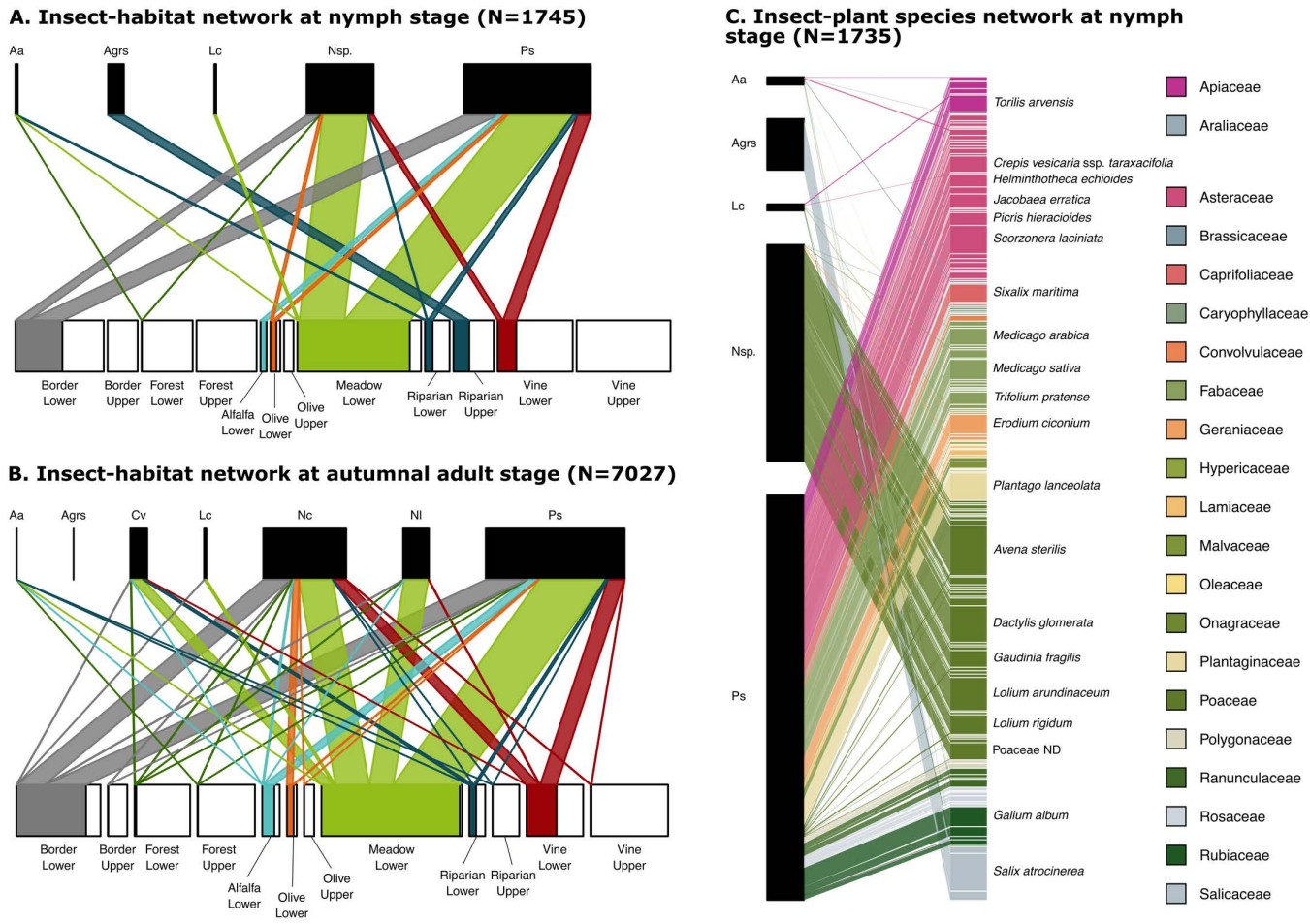

**Fig 2. Insect-habitat networks for nymphs (A), for autumnal adults (B), and insect-plant species networks for nymphs (C).**

At the resource (bottom) level of each network in A and B, the width of each habitat box (filled plus empty portion) is proportional to its sample size (i.e. number of subsites among all subsites surveyed) in the network displayed. For each habitat, the empty portion of the box shows the proportion of subsites on which no insect was found. Insect species are abbreviated as follows Aa: *Aphrophora alni*, Agrs: *Aphrophora* grp. *salicina*, Cv: *Cicadella viridis*, Lc: *Lepyronia coleoptrata*, Nc: *Neophilaenus campestris*, Nl: *Neophilaenus lineatus*, Nsp.: *Neophilaenus* sp. And Ps: *Philaenus spumarius*. In panel C, only the names of the most abundant plant species are displayed for the sake of readability. Note that the difference between panels A and C in the number of nymphs is due to 10 nymphs found on plants that could not be identified even at the family level. See S6 Appendix for details on the computation involved for insect-habitat networks. See S4 Appendix for insect-habitat network details per region, and S7 Appendix for insect-plant species network details per region.

## Habitat exploitation by xylem feeders

The total abundance of xylem feeders per subsite was significantly correlated with the habitat type (Fig 3, S8 Appendix), meaning that habitats were not equally exploited by insects (Fig 2). The least exploited habitats, for both xylem feeder abundance and diversity, were the upper strata of forests, vines, riparian areas, borders (i.e. hedges), olive groves, as

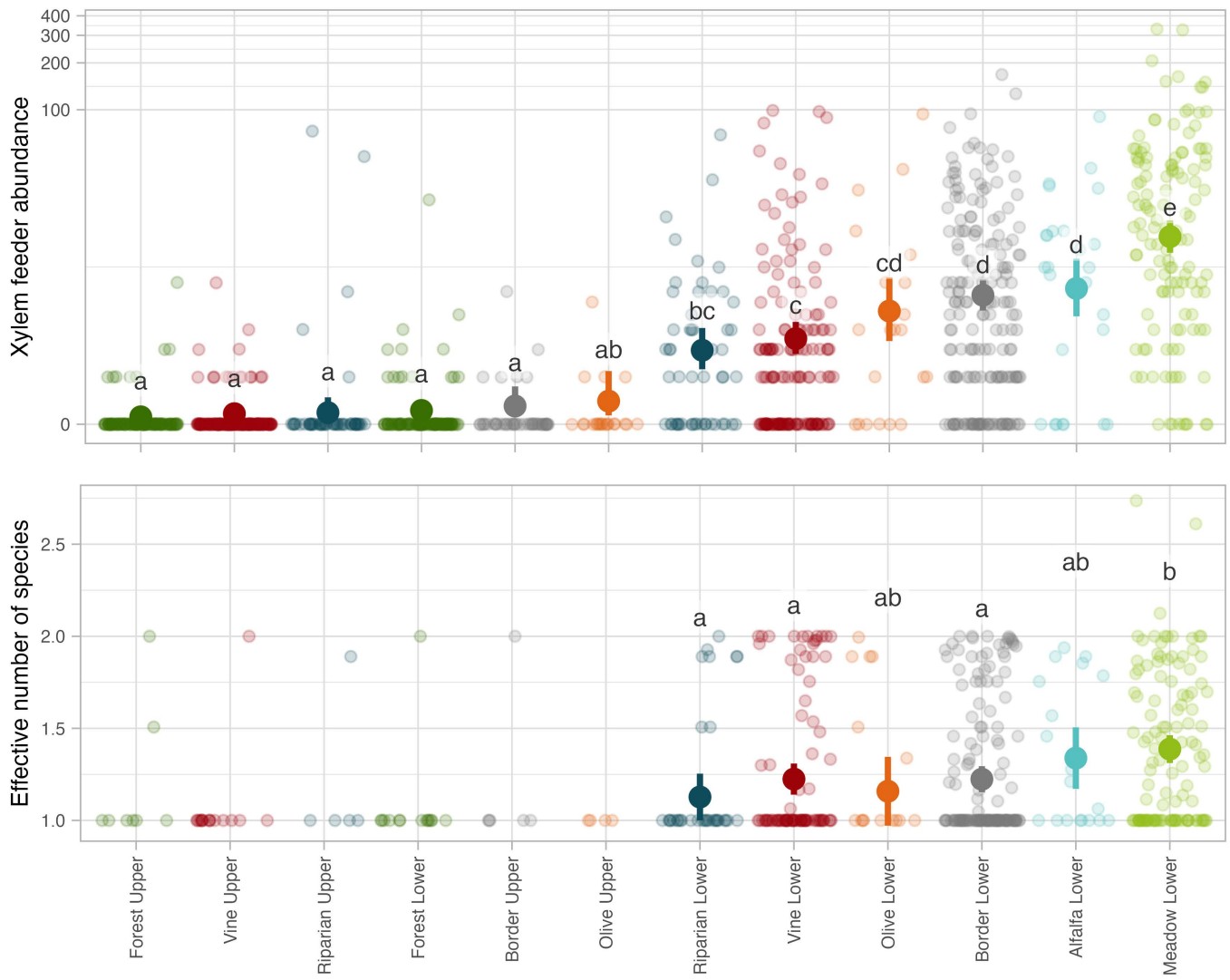

**Fig 3. Xylem feeder abundance and diversity for each habitat.** (A) Abundance of xylem feeders for each habitat. (B) Xylem feeder species diversity (effective number of species, exponential of Shannon index) for each habitat. Letters depict the significance of the effect of habitats and regions. For each panel taken independently, modalities sharing a letter do not differ significantly. Details per region can be found in S9 Appendix.

well as the lower stratum of forests (Fig 3, and see empty boxes in Fig 2). Due to lack of variation of insect diversity in these habitats (always only one xylem feeder species found), xylem feeder diversity could not be statistically compared to other habitats. Xylem feeder abundance was intermediate in lower strata of riparian areas, vines, olive groves, borders and alfalfa. Finally, the habitat where xylem feeders were the most abundant was the lower stratum of meadows (Fig 3, see also the proportion of full boxes in Fig 2). In terms of diversity, although a significant effect of habitat was detected (S5 Appendix), most habitats shared similar diversities (Fig 3).

Overall, xylem feeder abundance per subsite was higher in the NAQ and PACA regions than in the OCC region, and diversity was higher in the PACA region than in the two others (S9 Appendix).

## Xylem feeders specialization on habitats and host plants

At the level of habitats, xylem feeders had similar specialization degrees (Figs 2 and 4A): most species were ubiquitous except for *A.* grp. *salicina* that was restricted to riparian habitats (Fig 2). Among all specialization metrics, this latter species was more specialized than *N. campestris*. Additionally, *N. lineatus* was mostly restricted to meadows, and globally more specialized than its congeneric *N. campestris*, and also more specialized than *P. spumarius*. At the habitat level, *Neophilaenus* sp. nymphs were distributed similarly to autumnal adults of *N. campestris* or *N. lineatus* (Fig 4).

Although ubiquitous – i.e. able to thrive in various habitats – the generalist xylem feeders had habitat preferences, and this was particularly clear for *P. spumarius*, *Neophilaenus* sp. and *C. viridis* (Fig 4B, S5 Appendix). Philaenus *spumarius* was found in higher abundances on lower strata of meadows and alfalfa fields (Fig 4B, S8 Appendix). It was found gradually less on the lower strata of field borders, olive groves, grapevines and riparian vegetation (S8 Appendix) and only rarely on the upper strata of field borders (i.e. hedges), grapevines, riparian zones, forests, olive groves, as well as on the lower stratum of forests. It was globally less abundant in the OCC than in the PACA or NAQ regions (S8 Appendix). *Neophilaenus* species (including here *N. campestris* and *N. lineatus* at both nymph and autumnal adult stages) were most abundant in meadows, and then, to a lesser extent, on the lower strata of the borders of vineyards/olive groves, of olive groves, of vineyards and of alfalfa (Fig 4B, S8 Appendix). Similarly to *P. spumarius*, *Neophilaenus* species were scarce on the upper strata of borders of vineyards/olive groves (i.e. hedges), of vineyards, of riparian zones, of forests, of olive groves, as well as on the lower stratum of forests and riparian areas. *Neophilaenus* species were more abundant in the PACA region than in the OCC or NAQ regions (S8 Appendix). Finally, *C. viridis* was found mostly on the lower stratum of riparian zones, then on meadows, and was rare on lower strata of alfalfa, of borders of vineyards/olive groves, vineyards and forests (Fig 4B). It was not found in the other habitats. *Cicadella viridis* was on average 6.5 times more abundant in the NAQ than in the PACA region (S5 Appendix) and it was not found in the OCC region.

*Philaenus spumarius* was found on 18 plant families, followed by *A. alni* (8), *L. coleoptrata* (6), *Neophilaenus* sp. (4), and *A.* grp. *salicina* (2; S4 and S7 Appendices). Two groups of species could be differentiated: *P. spumarius*, *L. coleoptrata* and *A. alni* were the most generalists, while *Neophilaenus* sp. and *A.* grp. *salicina* were the most specialists (Figs 2C and 4A). *Neophilaenus* sp. was almost restricted to Poaceae and *A.* grp. *salicina* was mostly restricted to Salicaceae (found only once on *Fraxinus excelsior*, an Oleaceae, Fig 2C).

At the plant species level, it was again *P. spumarius* that was found on the highest number of plants (87 species), followed by *Neophilaenus* sp. (31), *A. alni* (10), *L. coleoptrata* (7), and *A.* grp. *salicina* (3, S7 Appendix). Consistently, all specialization metrics combined, *P. spumarius* was more generalist than *A.* grp. *salicina*, and the three other taxa – *Neophilaenus* sp., *A. alni* and *L. coleoptrata* – had intermediate specialization levels (Fig 3A).

## *Xf* prevalence in xylem feeders

Overall 714 *P. spumarius*, 156 *N. campestris*, 96 *N. lineatus*, 31 *C. viridis*, 17 *L. coleoptrata* and 3 *A. alni* sampled in the NAQ (n = 865) or PACA (n = 152) regions were screened for *Xf* (S1 Appendix). All insects tested negative.

## Discussion

In the present study, we identified insect-habitat-plant interaction networks in vineyards and their surrounding landscapes. We also assessed the prevalence of *Xf* in the environment based on a sentinel insect approach.

### Herbaceous semi-natural habitats are the main sources of xylem feeders

Our first hypothesis was supported by the results. The habitats where xylem feeders are the most abundant are indeed mostly semi-natural habitats. Exploring a large diversity of habitats of the agricultural mosaic, we found two groups of habitats. The first group gathers habitats that are almost free of xylem feeders. It includes the upper strata of forests, of

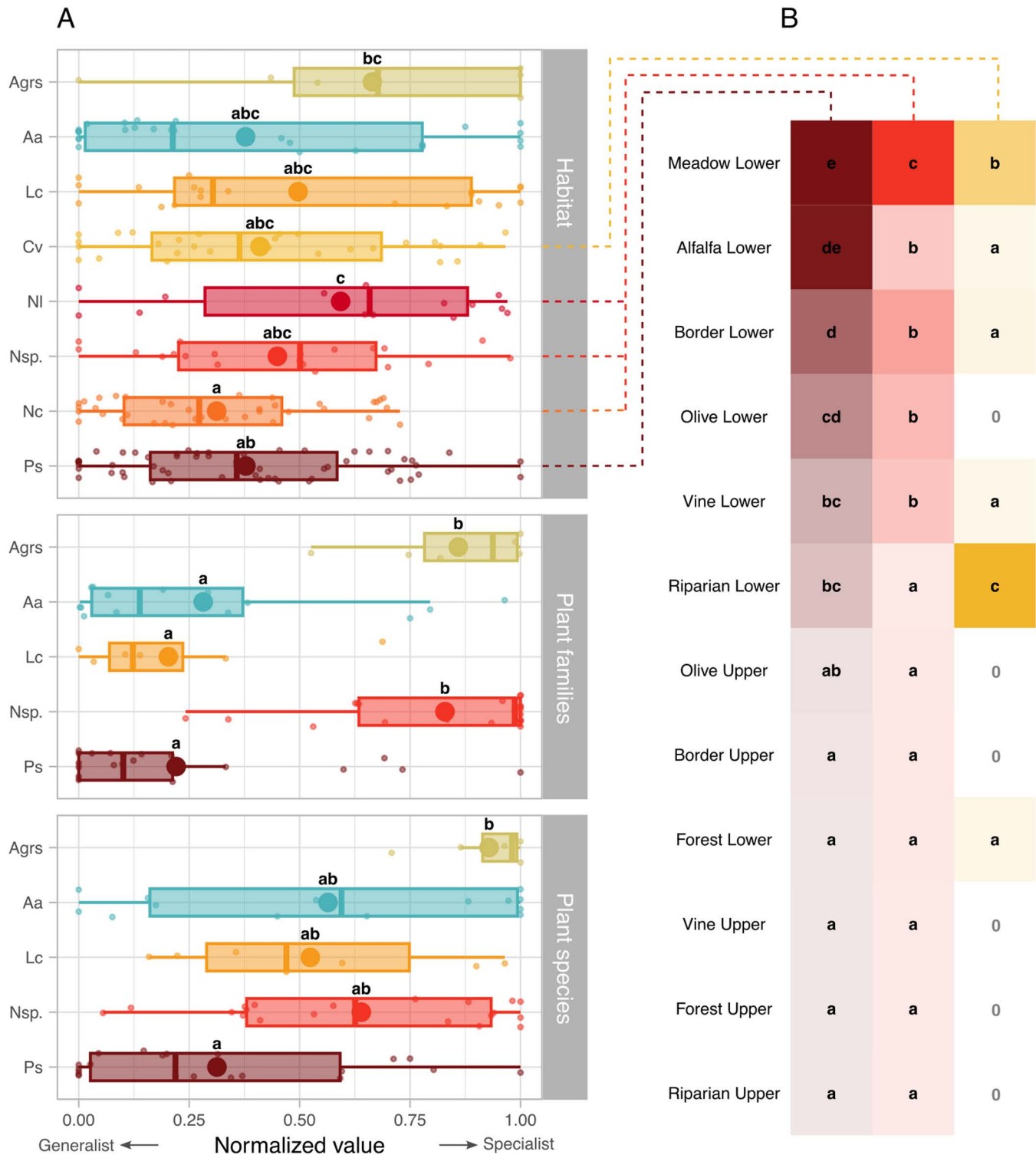

**Fig 4. Insect-habitat-plant specialization and habitat preferences of xylem feeders.** (A) Distribution of specialization values for insect-habitat (top panel), insect-plant families (middle panel) and insect-plant species (bottom panel) networks. Small points in the background depict the data: one point is one specialization metric for a given species in each network (e.g. one point in the top panel depicts the resource range of *P. spumarius* in the insect-habitat network assessed in the NAQ region in fall 2020). The big points depict means. The boxplots depict quartiles and extend up to 1.5 times the inter-quartile range. The letters report statistics performed on the distributions: in each panel independently, species that share a letter have

specialization degrees that are not significantly different. Note that in the top panel, habitat preference for *Cicadella viridis*, *Neophilaenus lineatus* and *Neophilaenus campestris* were only assessed at the autumnal adult stage and that of *Neophilaenus* sp. was only assessed at the nymphal stage. (B) Habitat preferences of the three most abundant xylem feeders. Color ramps are scaled independently for the three species and depict log-transformed estimated marginal means of GLMMs. For each species independently, habitats sharing a letter do not differ significantly. Zeros indicate habitats where the species was never collected. Insect species are abbreviated as follows Aa: *Aphrophora alni*, Agrs: *Aphrophora* grp. *Salicina*, Cv: *Cicadella viridis*, Lc: *Lepyronia coleoptrata*, Nc: *Neophilaenus campestris*, Nl: *Neophilaenus lineatus*, Nsp.: *Neophilaenus* sp. And Ps: *Philaenus spumarius*. See S5 and 10 Appendices for more details.

riparian vegetation, of vineyards, of olive groves, hedges, and the lower stratum of forests. This is a reassuring result for forests because some tree species, *Quercus* species in particular, are vulnerable to *Xf* [78]. The second group comprises habitats substantially exposed to xylem feeders, with an increasing gradient from the lower stratum of riparian vegetation up to meadows (very exposed).

For the two crops of significant economic value included in our sampling scheme, grapevine and olive, xylem feeders were found very rarely on the foliage. Two habitats that host large populations of xylem feeders are nevertheless near grapevine or olive foliage: the lower strata of vineyard/olive grove and their herbaceous borders, especially in PACA vineyards. This is in line with several studies showing that most potential vectors of *Xf* reach high abundances in herbaceous habitats – see e.g. Rodrigues et al. [79] for a recent study in Portuguese vineyards – especially at the nymphal stage [80]. Xylem feeders have been reported to migrate to crop foliage when the herbaceous vegetation gets too dry in the context of Italian olive groves [19,81], but also in Californian vineyards [82]. Besides inter-rows and borders, meadows and alfalfa fields, many of which were within the dispersal ability of *Xf* vectors to vineyards studied here, hosted the largest xylem feeder communities. Similarly, in almond orchards and associated landscapes in California, xylem feeders are extremely abundant in open herbaceous habitats that contain many host species for *Xf* [83,84] and insect transfer from these habitats to almond nurseries were suggested as a potential infection pathway to almond trees [83]. In our context, xylem feeders are also likely to move from alfalfa or meadows to vineyards or olive groves at the hay harvest. Indeed, migration to neighboring cereal crops following hay cutting have been reported for *P. spumarius* in Ohio [32] or migration to wheat fields has been observed in the NAQ region (XM, personal observation, 2023). In summary, across these different contexts (Italian olive groves, Californian vineyards, Californian almond groves), there is a consistent pattern where xylem feeder communities establish themselves on herbaceous covers and move into foliage when the herbaceous habitat conditions deteriorate (i.e. drying or harvesting). Our results suggest that this pattern could lead to *Xf* spread to vineyards (and olive groves) of southern mainland France as well. However, what we currently lack is year-round monitoring of xylem feeders to detect potential migrations to crop foliage after summer droughts or harvesting operations.

In this context, we identify few options to lower the transmission risk. The best option still remains to increase surveillance to avoid introduction and spread of the subspecies fastidiosa in France. There is little room to lower risk through sowing of inter-rows and of headlands, as many potential vectors of *Xf* are highly generalist. Still we could advise favoring dicotyledons to limit *Neophilaenus* spp. populations. *Avena* spp. that were seen in some vineyards in association with *Vicia faba* do not seem advisable as they host considerable populations of *Neophilaenus* spp. Soil tillage to reduce vegetation cover could be the most efficient lever [85] but this practice favors soil erosion especially in Mediterranean areas [86]. At the landscape scale, a stepwise harvesting of meadows and alfalfa plots might be interesting to avoid massive migrations of xylem feeders.

Interestingly, two contrasted dissemination pathways reported i/ in Californian vineyards and *Citrus* groves or ii/ in Corsican groves are very unlikely in the context studied here. The former involves *G. atropunctata* which is considered responsible for most of the primary transmission from riparian habitats to grapevines [27,87] and then *Homalodisca vitripennis* for most of the secondary transmission as it reaches high abundances within vineyards and *Citrus* groves [5]. In our context, *A.* grp. *salicina* or *C. viridis* are unlikely to play the role of *G. atropunctata* because *A.* grp. *salicina* is restricted to Salicaceae in riparian habitats and because *C. viridis* appears almost unable to transmit *Xf* [88,89].

Furthermore, in our context, xylem feeders do not reach high densities within the crops, unlike the situation observed with *H. vitripennis.* The Corsican example involves *P. spumarius* that shows a very strong preference for *C. monspeliensis* in olive and *Citrus* grove landscapes [24]. A potential dissemination pathway was identified at the interface between *C. monspeliensis* scrubland and groves. Here, woody habitats were not identified as significant sources of xylem feeders. Apart from riparian habitats mentioned earlier, we did not find any xylem feeder likely to spread Xf at the forest-agroecosystem interface or within forests.

## Most xylem feeders are generalists

First note that our analysis of insect preferences is based on two sampling periods whereas they show three pronounced abundance peaks per year (S2 Appendix): one in the spring for nymphs (sampled), one in the spring for adults (not sampled), and one in the autumn for adults (sampled). Therefore our discussion only applies to these two periods of their lives, and our results on adults cannot be generalized to their whole adult life. Indeed, habitat or feeding preferences can change markedly during the year [31], and further studies will be needed to assess the preferences of xylem feeders in mid-May in France.

Our second hypothesis was that *P. spumarius* would be extremely generalist and that other xylem feeders would be mostly specialists at the habitat and host plant levels. Using interaction networks that have previously proven effective in assessing habitat specialization in insects [64,90], we found that this hypothesis was only partially supported. At the habitat level, the degree of specialization of *P. spumarius* was very similar to that of *A. alni*, *C. viridis*, *L. coleoptrata* and *N. campestris*. This suggests that the widely reported ubiquity of *P. spumarius* [22,29] is shared with most spittlebugs and sharpshooters in the studied landscapes of southern France. Despite this ubiquity, xylem feeders did not distribute evenly among all studied habitats. As expected from the literature, *Neophilaenus* spp. preferred meadows over other habitats. *Neophilaenus lineatus* was indeed almost restricted to meadows. *Philaenus spumarius* also showed a marked preference for meadows. It is noteworthy that we confirm for the first time in Europe its association with alfalfa. This association was only documented in Ohio and Michigan so far [32,91]. Finally, *A.* grp. *salicina* was almost restricted to riparian habitats.

The plant family level (assessed at the nymph stage only) was more relevant to identify and differentiate specialization level among spittlebugs. *Philaenus spumarius*, *A. alni* and *L. coleoptrata* were generalists. On the opposite, *Neophilaenus* sp. and *A.* grp. *salicina* were specialists, each thriving on almost only one plant family, confirming the biological informations available for these species [28,39].

Finally, although *P. spumarius* had the highest generality score at the plant species level, it was not significantly more generalist than *A. alni*, *L. coleoptrata* or *Neophilaenus* sp. The 18 plant families on which it was found in the present study were also reported by Thompson et al. [40]. However, we observed *P. spumarius* on 6 plant genera not yet reported in the literature – namely *Anacyclus*, *Carthamus*, *Cota*, *Petrorhagia*, *Thrincia* and *Thyrimnus* – and 30 new plant species records (S11 Appendix). When we group all association records for *P. spumarius* and following the GBIF Backbone Taxonomy [55], we reach a total of 670 plant genera and 1279 plant species (S11 Appendix).

Interestingly, species in *Neophilaenus* were extremely specialized at the plant family level: they were found only occasionally on dicots. They were relatively generalists at the plant species level, being able to feed on a wide variety of graminaceous species. As reported in previous studies, *Neophilaenus* spp. were frequently found on *Dactylis glomerata* and *Avena sterilis* [39,80,92] but we also highlight high frequencies on *Lolium arundinaceum*, *Hordeum vulgare*, *Lolium rigidum* or *Gaudinia fragilis*. Our study provides a list of 31 species of Poaceae on which nymphs of *Neophilaenus* spp. can feed, 26 of which are new when we compare our list with the 21 host species reported by Antonatos et al. [92].

Finally, we found several nymphs of *A. alni* on *Pteridium aquilinum* (opportunistic extra samplings, S12 Appendix), one of the most ubiquitous species in the world, able to develop below forest canopies and in open areas [93]. More work is needed to determine if *A. alni* only occasionally feeds on that fern, or if bracken covers represent a significant reservoir of this xylem feeder.

To summarize the answer to our first research question, we expected *P. spumarius* to stand out clearly as the most generalist [29,40] at all studied levels, however this was generally not the case. We hypothesize that this expectation was driven by the frequent confusion between abundance and generalism [62,94] and the relatively poor knowledge we have on other xylem feeders. Here we tried to disentangle dietary niche from abundance by using several tools to better describe our networks [63,68]. Our results indicate that *P. spumarius* is abundant and generalist. *Aphrophora alni*, *L. cole-optrata* – and to a lesser extent *C. viridis* – are also generalists, but not abundant. The only "true" specialist in our study, *A.* grp. *salicina*, is generally not abundant. Finally, *Neophilaenus* has a hybrid status, being extremely specialist at the plant family level, but generalist at the habitat and plant species levels. These results are in line with the general pattern described by Fort et al. [94] in plant-pollinator and plant-seed dispersers networks: abundance implies generalism (i.e., specialists are rarely abundant), whereas the reverse is false (i.e. generalists can be rare).

A limitation of our study is that we cannot rigorously disentangle which of the plant (species, family) or habitat specialization determines the other. The results observed for *A.* grp *salicina* and *Neophilaenus* spp. suggest that for these two taxa, plant species and family respectively determine their habitat specialization. Further studies could try to disentangle these specialization levels especially for generalist xylem feeders. A perspective would be to conduct independent botanical surveys, and test whether or not generalist xylem feeders are found on plants proportionally to their soil coverage, as done for instance in plant-pollinator networks [95].

### *Xf* was not detected in the insects sampled

*Xf* was not found in the xylem feeders analyzed in this study. The absence of *Xf* in our samples from the PACA region is surprising, because *Xf* foci were officially detected ca. 25 km from some of our sampling sites in 2015 [96], and *Xf* dissemination by insects in Apulia is reported at about 10 km per year [97], though in different landscape conditions. This result suggests that the spread of *Xf* is low in this area, possibly due to the eradication of infected plants, possibly also because infected plants were found in urban environments that have few large herbaceous areas favorable to xylem feeders. The bacterium was not present in sampled agricultural areas and the hazard of *Xf* transmission to plants is therefore very limited to date, even though xylem feeders are present. The absence of *Xf* in the NAQ region is less surprising because *Xylella* had only been found once in vine bark, a detection that has not been confirmed ever since. This result suggests that one of the main wine-growing areas in France is still uninfected by the *Xf*. The closest *Xf* focus to date is more than 100 km away (near Toulouse [98], ssp. *multiplex*). However the recent finding of *Xf fastidiosa*, the subspecies that is harmful to vineyards, progressing quickly in southern Italy [12,13] indicates that it may become a threat for the European mainland. In both areas as well as in in the French Occitanie region, we recommend enhancing surveillance and supplementing it with the sentinel insect approach as a powerful tool for the early detection of infection foci, especially when plants do not show clear symptoms [43].

Available species distribution models indicate that the current climate conditions are suitable for *Xf fastidiosa* in different territories of Europe, including the French regions surveyed in the present study [99]. Climate suitability is also expected to increase in the coming decades as climate change will lead to milder winters, hence lowering the cold curing effect [17,100]. These trends indicate that the risk of *Xf fastidiosa* establishment and/or expansion in Europe is and will remain high. Less is known about the impact of climate change on the range of the insect vectors, but Godefroid et al [101] showed that *P. spumarius*, the main vector of *Xf,* is expected to remain present on most of the continent. According to these results, both *Xf fastidiosa* and its main insect vector will remain a serious risk in most of the wine-producing regions in Europe. Importantly, surveillance should also target quarantine, highly competent vectors of Xf especially *H. vitripennis* for which the climate conditions of the European continent are and will remain suitable [102] or *G. atropunctata*.

Finally, it seems crucial to develop and test prophylactic and control strategies in the near term, particularly since *Xf fastidiosa* is currently absent. This will allow us to experiment with several strategies on xylem feeders communities without any epidemiological repercussions.

## Supporting information

**S1 Appendix. Geographical distribution of the sample size.**
(ZIP)

**S2. Striking image.**
(PNG)

## Acknowledgments

We thank Jérémy Minguez, Malika Rouzes, Étienne Ramadier, Anna Dessaudes, Jennifer Dudit, Alice Bedani and Julien Pradel for their punctual help in the field work. We thank the UMR SAVE, especially Pauline Tolle and Adrien Rusch for giving us access to the BACCHUS winegrower network. We are grateful to the anonymous reviewer for its feedback on this work.

## Author contributions

**Conceptualization:** Xavier Mesmin, Marguerite Chartois, Pauline Farigoule, Jean-Yves Rasplus, Astrid Cruaud, Jean-Pierre Rossi.

**Data curation:** Xavier Mesmin.

**Formal analysis:** Xavier Mesmin, Marguerite Chartois.

**Funding acquisition:** Jean-Pierre Rossi.

**Investigation:** Xavier Mesmin, Marguerite Chartois, Pauline Farigoule, Christian Burban, Jean-Claude Streito, Jean-Marc Thuillier, Éric Pierre, Maxime Lambert, Yannick Mellerin, Olivier Bonnard, Inge van Halder, Guillaume Fried.

**Methodology:** Xavier Mesmin, Marguerite Chartois, Jean-Yves Rasplus, Astrid Cruaud.

**Project administration:** Astrid Cruaud, Jean-Pierre Rossi.

**Supervision:** Astrid Cruaud, Jean-Pierre Rossi.

**Validation:** Jean-Claude Streito, Guillaume Fried.

**Visualization:** Xavier Mesmin, Marguerite Chartois.

**Writing – original draft:** Xavier Mesmin, Marguerite Chartois.

**Writing – review & editing:** Christian Burban, Inge van Halder, Astrid Cruaud, Jean-Pierre Rossi.

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
