## [Decision Letter · Decision Letter 0]

31 Mar 2025

*Xylella fastidiosa*

Dear Dr. Mesmin,

Thank you for submitting your manuscript to PLOS ONE. After careful consideration, we feel that it has merit but does not fully meet PLOS ONE’s publication criteria as it currently stands. Therefore, we invite you to submit a revised version of the manuscript that addresses the points raised during the review process.

We look forward to receiving your revised manuscript.

Kind regards,

Tzen-Yuh Chiang

Academic Editor

PLOS ONE

Journal Requirements:

3. We note that Figure 1 in your submission contain [map/satellite] images which may be copyrighted. All PLOS content is published under the Creative Commons Attribution License (CC BY 4.0), which means that the manuscript, images, and Supporting Information files will be freely available online, and any third party is permitted to access, download, copy, distribute, and use these materials in any way, even commercially, with proper attribution. For these reasons, we cannot publish previously copyrighted maps or satellite images created using proprietary data, such as Google software (Google Maps, Street View, and Earth). For more information, see our copyright guidelines: http://journals.plos.org/plosone/s/licenses-and-copyright .

Reviewers' comments:

Reviewer's Responses to Questions

**Comments to the Author**

1. Is the manuscript technically sound, and do the data support the conclusions?

Reviewer #1: Partly

2. Has the statistical analysis been performed appropriately and rigorously?

Reviewer #1: I Don't Know

3. Have the authors made all data underlying the findings in their manuscript fully available?

Reviewer #1: No

4. Is the manuscript presented in an intelligible fashion and written in standard English?

Reviewer #1: Yes

Reviewer #1: In this study, the authors examined insect-habitat and insect-plant networks in three French wine-growing regions to identify xylem feeder habitats and specialization. Surveys across 92 landscapes showed meadows and alfalfa fields hosted the largest communities, while grapevines had few. Philaenus spumarius, Aphrophora alni, Lepyronia coleoptrata, and Cicadella viridis were generalists, while Aphrophora grp. salicina was a strict specialist on riparian forests and Salicaceae. Screening 1,017 insects found no Xylella fastidiosa, suggesting it is not widespread, while providing key insights into potential vector ecology. The study is both innovative and valuable, particularly due to its extensive site coverage and novel sampling design. A key strength is the simultaneous collection of insect-habitat and insect-plant network data alongside Xylella fastidiosa (Xf) screening, providing a comprehensive ecological perspective. However, several major concerns need to be addressed.

Clarity: The manuscript requires improvement in multiple sections, especially in the descriptions of network and statistical analyses. Additionally, the Introduction should better contextualize the study within existing research.

Sampling: A significant limitation is that sites were sampled only once, and adult insects were monitored exclusively in autumn, which may impact the completeness of the dataset and trend interpretation (see specific comments below).

Analysis: The analytical approach is difficult to follow, and a clearer explanation is needed to ensure full transparency for the reader. Additionally, potential methodological issues should be addressed, including the confounding effects of insect species and habitat characteristics in specialization analyses, as well as concerns regarding model construction and simplification procedures.

#####

Introduction. Interesting and well written. Besides minor comments, I would suggest the authors to orderly present the different Xf subspecies, their main problematics in terms of plant protection, their main vectors and relative characteristics. I think this would help a non-specialized audience to better follow the study.

L59 New reference: Cornara, D., Boscia, D., D’Attoma, G., Digiaro, M., Ligorio, A. M., Loconsole, G., ... & Saponari, M. (2024). An integrated strategy for pathogen surveillance unveiled Xylella fastidiosa ST1 outbreak in hidden agricultural compartments in the Apulia region (Southern Italy). European Journal of Plant Pathology, 1-9.

L67 and 82-84 Authors should better highlight the previous studies that adopted similar approach (species-habitat network) for Xf vectors in Europe (e.g., Cappellari et al. 2022). So far, Cappellari et al. (2022) is only mention to support the statement “Large populations of P. spumarius were also reported in grasslands”, which is reductive. For example, in their study Cappellari et al. found that Philaenus spumarius was mostly collected in olive groves but it acted as a super-generalist species, occupying all the habitats (this seems to match some of the authors’ findings). I suggest the authors to better highlight the innovative aspects of their work on the base of past studies.

I would suggest authors to more clearly present the different Xf subspecies, their relevance for crop protection (which plant species might be strongly affected?) and the relative main vectors and their characteristics (as shortly done at L97-101). Authors could also briefly present the known distribution for the main Xf vetcors in Europe. This work will be very interesting for Xf specialists but it could also appeal a broader, less knowledgeable audience.

L105 New reference: Cornara, D., Boscia, D., D’Attoma, G., Digiaro, M., Ligorio, A. M., Loconsole, G., ... & Saponari, M. (2024). An integrated strategy for pathogen surveillance unveiled Xylella fastidiosa ST1 outbreak in hidden agricultural compartments in the Apulia region (Southern Italy). European Journal of Plant Pathology, 1-9.

#####

Materials and Methods: This section requires significant improvement in clarity and completeness. Additional details should be provided to enhance transparency, ensuring that the methodology can be fully understood and replicated. Specific aspects, such as sampling procedures, data collection protocols, and analytical methods, should be described more precisely to strengthen the study’s rigor.

L130 Authors should indicate which type of habitats were considered and how different patches were selected (one habitat type per buffer? based on relative abundance of each habitat?).

L148-150 I am concerned about this point. Authors sampled adults in two years, only in autumn, indicating that unpublished phenological data in the South and in the West of France show peak densities of adults in the autumn. First, I think authors should more strongly support this statement. Adult abundance peak often occurs in spring (Bodino et al. 2019, Hasbroucq et al. 2020, Cappellari et al. 2022) although high densities have been observed later along the season (Bodino et al. 2021). Can authors present in the supplementary (very aggregated) data on vector phenology in France, in order to support their claims (without losing the possibility of future publications)? Again I think this is important, because, if this is not the case, authors might have missed the real population peak. Second, with only one season sampled, authors cannot be sure about the vector habitat preferences. In fact, these vectors are mobile, and they might use habitat differently in different seasons. For example, Cappellari et al. 2022 shows that grasslands are preferred by P spumarius adults disproportionally more in spring than in autumn. In other words, authors can argue about habitat preference in autumn, and we don’t know what happen in spring (generally a very important season for these vectors, when actual dispersion may occur). Finally, although the total number of sites is high, each subsite in each site was sampled only once (L160): this type of snapshot sampling does not allow to properly capture local variability, increasing uncertainty. I think authors should better justify their choice and acknowledge the potential limitation of the study.

Bodino, N., Cavalieri, V., Pegoraro, M., Altamura, G., Canuto, F., Zicca, S., ... & Bosco, D. (2021). Temporal dynamics of the transmission of Xylella fastidiosa subsp. pauca by Philaenus spumarius to olive plants. Entomologia Generalis, 1-17.

Bodino, N., Demichelis, S., Simonetto, A., Volani, S., Saladini, M. A., Gilioli, G., & Bosco, D. (2021). Phenology, seasonal abundance, and host-plant association of spittlebugs (Hemiptera: Aphrophoridae) in vineyards of Northwestern Italy. Insects, 12(11), 1012.

Hasbroucq, S., Casarin, N., Czwienczek, E., Bragard, C., & Grégoire, J. C. (2020). Distribution, adult phenology and life history traits of potential insect vectors of Xylella fastidiosa in Belgium. Belg. J. Entomol, 92, 1-21.

L150-151I don’t think that changing buffers each year completely solve the problem of pseudoreplication. The site selection as described so far does not exclude spatial/temporal dependence among sites. First, new buffers should have been selected to a sufficient distance to present and past buffers (2km distance between two centroids) in order to be confident that different vector populations were sampled. Also, the effect of “spece” and “time” is now confounded (differences among years depend on time or on different characteristics of sites sampled?).

L160 subsite here coincide habitat right?

L163-164 not clear, I suggest rephrasing.

L167-174 Please provide references to support this approach. Changing quadrat area should theoretically allow to weight vector abundance on vegetation abundance. However it is not clear to me if this was done systematically and consistently across habitat (stratus by site). How authors would interpret the same amount of vectors in e.g., upper strata of vineyards vs. lower strata of meadows? Please elaborate on this (and provide references).

L 187 So, in the selected sites, were all P. spumarius and N. campestris captured also tested?

Network analysis. I found the description of network analysis quite confusing and difficult to follow. As mentioned below, I suggest the authors to first explain the overall approach (e.g., it could be useful to mention at the beginning that authors want to separately explore insect-habitat, insect-plant families and insect-plant species networks). Then authors should clearly state for each type of analysis which data were used (adults vs nymphs) the resolution/aggregation of the data (how many networks?), the type of metrics extracted. Please, considering the complexity of the study (in terms of structure) and analyses (in terms of different type of aggregation/metrics used), I strongly suggest the authors to help the reader briefly explaining why they selected that particular metric/strategy and how it works (e.g., Lami et al. 2020 did a very good job in breaking down the analytical process and explaining each metric used). Also, I have another doubt regarding the analyses: authors combine two types of network: species habitat and species-plant networks. My doubt is that since constructing insect-plant interaction networks from data collected across different habitats, there is a risk of confounding the effect of insect species with the effect of habitat characteristics, particularly due to the varying plant communities across habitats. This might lead to misinterpretation of dietary specialization, making it difficult to determine whether an insect’s observed feeding behavior is due to intrinsic dietary preferences or simply a reflection of the available plant species in a given habitat. Did authors consider this? In order to disentangle these effects authors might 1) Standardize plant availability across habitats. I think authors attempted to do this, even thought there were still big differences in terms of plant cover across habitats (see my comment above). Also, I don’t think this would solve the problem of different plant diversity across habitat. 2) Compare insect specialization across habitats. Does diet changes with plant availability (suggesting habitat-driven effects)?

L215-218 This is not clear. Were both nimphs and adults considered in this aggregation? Were all the three years aggregated or kept separated? If only 3 networks were generated, it seems like there were not enough replicates to properly analyse specialization. This section should be clearer. Also, are authors talking about species-habitat networks (I guess) or vector-plant network? I see now that this is explain below. I suggest the authors to begin this paragraph clearly presenting the analytical approach adopted here.

L220 something strange in “including in low connectance networks”.

L221 what is d’?

L22-224. Please, better explain this concept. Also are you sure that ref 53 used the same approach?

L231-232 I don’t understand what “(resp. plant family or species)” means. Also, why at 233 plants are mentioned when speaking about insect-habitat networks? Please make this part clearer.

L234-235 so 9 networks? Please make number of networks explicit.

L247 It is not clear to me why “metric” is a predictor here. Shouldn’t each metric be analyzed separately? I might have missed something but it seems to me like analyzing abundance, diversity, evenness in the same model. This would not make sense, so maybe I misunderstood. Please explain and provide appropriate explanation. Also, shouldn’t the model include the year as predictor (even if authors are not interested in the year effect)? I don’t understand what the random factor is…

L255-256 I don’t agree with the authors in using this approach. Removing insignificant fixed terms iteratively in GLMMs is a bad practice because it inflates Type I error, increases overfitting, and makes p-values unreliable. Instead of relying solely on statistical significance, models should be built based on ecological reasoning rather than just stepwise selection. A more robust approach is to retain biologically meaningful predictors even if not significant.

L263 Again, I would include the year as fixed factor. As it is, the model consider your data as collected in 700 independent sites, all collected the same year. This is not the case.

L265-266 Including a random factor with only two levels in a GLMM is problematic because it provides insufficient data to estimate random variance accurately, often leading to variance estimates close to zero. This can cause inflated Type I error rates, making fixed effect estimates appear more significant than they actually are. Instead of treating a two-level factor as random, it is better to model it as a fixed effect or ensure more levels are included for proper variance estimation.

Bolker, B. M., Brooks, M. E., Clark, C. J., Geange, S. W., Poulsen, J. R., Stevens, M. H. H., & White, J. S. S. (2009). Generalized linear mixed models: a practical guide for ecology and evolution. Trends in Ecology & Evolution, 24(3), 127-135. https://doi.org/10.1016/j.tree.2008.10.008

Gelman, A., & Hill, J. (2007). Data Analysis Using Regression and Multilevel/Hierarchical Models. Cambridge University Press.

Harrison, X. A., Donaldson, L., Correa‐Cano, M. E., Evans, J., Fisher, D. N., Goodwin, C. E., ... & Inger, R. (2018). A brief introduction to mixed effects modelling and multi‐model inference in ecology. PeerJ, 6, e4794. https://doi.org/10.7717/peerj.4794

Zuur, A. F., Ieno, E. N., Walker, N. J., Saveliev, A. A., & Smith, G. M. (2009). Mixed Effects Models and Extensions in Ecology with R. Springer Science & Business Media.

L269-270 Why did authors chose this metric? Wouldn’t be more straightforward to present species richness and evenness? Also, does “set to NA for null abundances” means absence of one species is not considered in the model? Please explain and provide support for your decisions.

#####

Results. Authors often present their results by different region. Is this really the core, interesting aspects of their results? Of course it has important management implications, but it seems to me not as relevant as generally study vector preferences and specialization in general. Also, this approach takes a lot of space in terms of figures (fig. 2,4). I would focus on general patterns and present the regional results in the text and in the supplementary.

General organization of analyses-result presentation. Authors first focus on species specialization and then general habitat preference (abundance by habitat). In my opinion, it would be more reasonable to first explore general effect of habitat on abundance/diversity and then explore specialization.

#####

Discussion generally reads well and it is well structured. I suggest the authors to better highlight the study limitations.

**Do you want your identity to be public for this peer review?** For information about this choice, including consent withdrawal, please see our Privacy Policy

Reviewer #1: No

---

## [Author Response · Author response to Decision Letter 1]

7 May 2025

[All responses to the editor and reviewer can be found in the uploaded document "Response to Reviewers.docx", with a more readable layout. They are also pasted below, our answers follow a ">" mark.]

PONE-D-24-38713

Insect-habitat-plant interaction networks provide guidelines to mitigate the risk of transmission of Xylella fastidiosa to grapevine in Southern France

Journal Requirements:

> We confirm that our manuscript meets PLOS ONE’s style requirements.

> No regulation applies to the collection of insect vectors (proven or putative) of Xylella fastidiosa in France. None of our sampling sites falls within a protected area and no restriction applied to insect collection in our sampling sites. Thus insect sampling did not require any permission from French authorities. Farmers were contacted before field visits through the network of our funder, Hennessy, and granted individually access to the fields.

A sentence was added in the M&M section to mention that.

> No copyrighted images were used to build the map in Fig 1. Country contours were displayed using Natural Earth (version 5.1.1) under a CC0 licence (https://www.naturalearthdata.com/about/terms-of-use/), and French departments were displayed using ADMIN EXPRESS database from the National Institute of Geography (IGN), freely available under a CC BY licence (https://www.data.gouv.fr/fr/datasets/admin-express-admin-express-cog-admin-express-cog-carto-admin-express-cog-carto-pe/#/community-reuses).

A simplified version of this sentence was added to Figure 1 caption, and to the captions of other maps in Appendices.

Comments to the Author

1. Is the manuscript technically sound, and do the data support the conclusions?

Reviewer #1: Partly

2. Has the statistical analysis been performed appropriately and rigorously?

Reviewer #1: I Don't Know

3. Have the authors made all data underlying the findings in their manuscript fully available?

Reviewer #1: No

> We are sorry that it was probably not clearly mentioned, but yes, all datasets and R codes used for data analyses and for computing figures were deposited on a public repository. Precisely, the dataset and R code (for both figures and statistics) were deposited on the 12th of May 2024 on Zenodo (https://zenodo.org/records/11181990). This is mentioned on the appropriate section “Statistical and data reporting” of the manuscript. This has been viewed 67 times and downloaded 36 times.

4. Is the manuscript presented in an intelligible fashion and written in standard English?

Reviewer #1: Yes  

Reviewers' comments: Review Comments to the Author

Reviewer #1:

In this study, the authors examined insect-habitat and insect-plant networks in three French wine-growing regions to identify xylem feeder habitats and specialization. Surveys across 92 landscapes showed meadows and alfalfa fields hosted the largest communities, while grapevines had few. Philaenus spumarius, Aphrophora alni, Lepyronia coleoptrata, and Cicadella viridis were generalists, while Aphrophora grp. salicina was a strict specialist on riparian forests and Salicaceae. Screening 1,017 insects found no Xylella fastidiosa, suggesting it is not widespread, while providing key insights into potential vector ecology. The study is both innovative and valuable, particularly due to its extensive site coverage and novel sampling design. A key strength is the simultaneous collection of insect-habitat and insect-plant network data alongside Xylella fastidiosa (Xf) screening, providing a comprehensive ecological perspective. However, several major concerns need to be addressed.

Clarity: The manuscript requires improvement in multiple sections, especially in the descriptions of network and statistical analyses. Additionally, the Introduction should better contextualize the study within existing research.

Sampling: A significant limitation is that sites were sampled only once, and adult insects were monitored exclusively in autumn, which may impact the completeness of the dataset and trend interpretation (see specific comments below).

Analysis: The analytical approach is difficult to follow, and a clearer explanation is needed to ensure full transparency for the reader. Additionally, potential methodological issues should be addressed, including the confounding effects of insect species and habitat characteristics in specialization analyses, as well as concerns regarding model construction and simplification procedures.

#####

Introduction. Interesting and well written. Besides minor comments, I would suggest the authors to orderly present the different Xf subspecies, their main problematics in terms of plant protection, their main vectors and relative characteristics. I think this would help a non-specialized audience to better follow the study.

> We strengthened the presentation of Xf subspecies, of their main vectors, and of the threats they pose for plant health.

L59 New reference: Cornara, D., Boscia, D., D’Attoma, G., Digiaro, M., Ligorio, A. M., Loconsole, G., ... & Saponari, M. (2024). An integrated strategy for pathogen surveillance unveiled Xylella fastidiosa ST1 outbreak in hidden agricultural compartments in the Apulia region (Southern Italy). European Journal of Plant Pathology, 1-9.

> We added this new reference.

L67 and 82-84 Authors should better highlight the previous studies that adopted similar approach (species-habitat network) for Xf vectors in Europe (e.g., Cappellari et al. 2022). So far, Cappellari et al. (2022) is only mention to support the statement “Large populations of P. spumarius were also reported in grasslands”, which is reductive. For example, in their study Cappellari et al. found that Philaenus spumarius was mostly collected in olive groves but it acted as a super-generalist species, occupying all the habitats (this seems to match some of the authors’ findings). I suggest the authors to better highlight the innovative aspects of their work on the base of past studies.

> We have more clearly described the super-generalist status of P. spumarius, both on host plants and habitats, as described in the results of previous studies (Bodino et al., 2019; Capellari et al., 2022; Villa et al., 2020), and we have also more clearly described what our study contributes to the subject.

I would suggest authors to more clearly present the different Xf subspecies, their relevance for crop protection (which plant species might be strongly affected?) and the relative main vectors and their characteristics (as shortly done at L97-101). Authors could also briefly present the known distribution for the main Xf vetcors in Europe. This work will be very interesting for Xf specialists but it could also appeal a broader, less knowledgeable audience.

> We added description of the five subspecies of Xf and their economic impacts, as well as more details on vectors (species and distribution).

L105 New reference: Cornara, D., Boscia, D., D’Attoma, G., Digiaro, M., Ligorio, A. M., Loconsole, G., ... & Saponari, M. (2024). An integrated strategy for pathogen surveillance unveiled Xylella fastidiosa ST1 outbreak in hidden agricultural compartments in the Apulia region (Southern Italy). European Journal of Plant Pathology, 1-9.

> We added this new reference.

#####

Materials and Methods: This section requires significant improvement in clarity and completeness. Additional details should be provided to enhance transparency, ensuring that the methodology can be fully understood and replicated. Specific aspects, such as sampling procedures, data collection protocols, and analytical methods, should be described more precisely to strengthen the study’s rigor.

> We thank the reviewer for these suggestions to improve the clarity of the methodology. We have implemented the suggested changes.

L130 Authors should indicate which type of habitats were considered and how different patches were selected (one habitat type per buffer? based on relative abundance of each habitat?).

> We provided clarifications about the distribution of land cover types within the buffer and the distribution of habitat types within the sites.

L148-150 I am concerned about this point. Authors sampled adults in two years, only in autumn, indicating that unpublished phenological data in the South and in the West of France show peak densities of adults in the autumn. First, I think authors should more strongly support this statement. Adult abundance peak often occurs in spring (Bodino et al. 2019, Hasbroucq et al. 2020, Cappellari et al. 2022) although high densities have been observed later along the season (Bodino et al. 2021). Can authors present in the supplementary (very aggregated) data on vector phenology in France, in order to support their claims (without losing the possibility of future publications)? Again I think this is important, because, if this is not the case, authors might have missed the real population peak.

> Thank you very much for raising that issue. We analyzed more thoroughfully our phenological data and we now show it in Appendix S2. We show two peaks of abundance in the year, one in mid-May / early June, and a second one in late September / October. As you mention, the highest densities were actually recorded in mid-May rather than in the autumn for aphrophorids as a whole. For P. spumarius alone, the peak also occurred in mid-May in OCC region, but two equivalent peaks were recorded in NAQ region, in mid-May and in late September.

We conclude that we sampled adults at a time that is meaningful in their population dynamic because they are very abundant. But we did not sample them at their highest densities, except for P. spumarius in NAQ region, as this species seems to display two similar peaks in the year in that place.

We changed the main text correspondingly. We state that sampling periods were pertinent regarding insect phenology, but that missing the peak density of adults (partly for logistic reasons, the financed project started in the summer 2020) is a limitation of our study.

Second, with only one season sampled, authors cannot be sure about the vector habitat preferences. In fact, these vectors are mobile, and they might use habitat differently in different seasons. For example, Cappellari et al. 2022 shows that grasslands are preferred by P. spumarius adults disproportionally more in spring than in autumn. In other words, authors can argue about habitat preference in autumn, and we don’t know what happen in spring (generally a very important season for these vectors, when actual dispersion may occur).

> Indeed. We are now more careful when talking about vector preferences. When mentioning adult preferences (for habitats) we reformulated for “autumnal adults preferences”. We also discuss this limitation in details at the beginning of the section "Most xylem feeders are generalists" of the Discussion.

Finally, although the total

---

## [Decision Letter · Decision Letter 1]

19 Jun 2025

*Xylella fastidiosa to grapevine in Southern France*
*PLOS ONE*

Dear Dr. Mesmin,

*Thank you for submitting your manuscript to PLOS ONE. After careful consideration, we feel that it has merit but does not fully meet PLOS ONE’s publication criteria as it currently stands. Therefore, we invite you to submit a revised version of the manuscript that addresses the points raised during the review process.*

**

We look forward to receiving your revised manuscript.

*Kind regards,*

*Tzen-Yuh Chiang*

Academic Editor

PLOS ONE

**

Reviewers' comments:

Reviewer's Responses to Questions

*
**Comments to the Author**
*

*Reviewer #1: (No Response)*

*2. Is the manuscript technically sound, and do the data support the conclusions??>*

*Reviewer #1: Yes*

*3. Has the statistical analysis been performed appropriately and rigorously? ?>*

*Reviewer #1: Yes*

*4. Have the authors made all data underlying the findings in their manuscript fully available??>*

*Reviewer #1: Yes*

*5. Is the manuscript presented in an intelligible fashion and written in standard English??>*

*Reviewer #1: Yes*

*Reviewer #1: I commend the authors for the substantial improvements made in this revised version. The manuscript is now significantly clearer, especially regarding the analytical framework and methodological structure. The authors’ willingness to engage constructively with reviewer feedback is evident and appreciated.*

However, the improved clarity has also brought to light a few remaining concerns, particularly regarding the statistical modeling and analytical choices:

1. Network Metrics and Replication

It is not entirely clear why network metrics were calculated at the regional level rather than at the buffer level (Lines 260–262 / 274–275). This approach reduces the number of replicates and may limit statistical power. Since buffers are spatially independent sampling units, calculating network metrics per buffer would seem more appropriate and in line with the hierarchical sampling design. If there is a rationale for discarding buffer-level replication, it should be clearly justified in the manuscript.

2. Random Effects Structure

The use of a crossed random effect structure rather than nesting sites within buffers is questionable. Sites are always nested within buffers, so a nested random structure (site within buffer) would more accurately reflect the sampling design. The current approach may misrepresent the true data structure and introduce biases.

3. Modeling Specialization Metrics

Including the network metric itself as a fixed effect within a single model to generalize across metrics is unconventional. Typically, each metric would be modeled separately to avoid inflating statistical power and maintain interpretability, as was done in the supplementary analyses. If the authors decide to retain their approach, they should clearly justify it and provide published references supporting this method.

*what does this mean? ). If published, this will include your full peer review and any attached files.*

**Do you want your identity to be public for this peer review?** For information about this choice, including consent withdrawal, please see our Privacy Policy

*Reviewer #1: No*

**

---

## [Author Response · Author response to Decision Letter 2]

24 Jul 2025

Reviewer #1: I commend the authors for the substantial improvements made in this revised version. The manuscript is now significantly clearer, especially regarding the analytical framework and methodological structure. The authors’ willingness to engage constructively with reviewer feedback is evident and appreciated.

-> Thank you for your appreciation. We believe that the suggested changes improved the manuscript.

However, the improved clarity has also brought to light a few remaining concerns, particularly regarding the statistical modeling and analytical choices:

1. Network Metrics and Replication

It is not entirely clear why network metrics were calculated at the regional level rather than at the buffer level (Lines 260–262 / 274–275). This approach reduces the number of replicates and may limit statistical power. Since buffers are spatially independent sampling units, calculating network metrics per buffer would seem more appropriate and in line with the hierarchical sampling design. If there is a rationale for discarding buffer-level replication, it should be clearly justified in the manuscript.

-> Thank you for raising this point. Indeed, we initially analyzed our data using networks computed at the buffer level, but the resulting networks were too small to be properly analyzed. We have added this information to the manuscript.

2. Random Effects Structure

The use of a crossed random effect structure rather than nesting sites within buffers is questionable. Sites are always nested within buffers, so a nested random structure (site within buffer) would more accurately reflect the sampling design. The current approach may misrepresent the true data structure and introduce biases.

-> We agree with your comment, and both the R code (updated on Zenodo) and the text in the M&M section were revised accordingly. Please note that this change did not affect the results presented, because the random effects were already nested by construction: site names were created by concatenating the buffer ID and the site number within each buffer.

3. Modeling Specialization Metrics

Including the network metric itself as a fixed effect within a single model to generalize across metrics is unconventional. Typically, each metric would be modeled separately to avoid inflating statistical power and maintain interpretability, as was done in the supplementary analyses. If the authors decide to retain their approach, they should clearly justify it and provide published references supporting this method.

-> We consider the different network metrics used to assess insect specialization as repeated measures of the same underlying phenomenon (i.e., specialization), applied to the same insects within the same networks. We deliberately selected metrics that vary in their sensitivity to network properties, ensuring that our conclusions are not overly dependent on a single metric.

To account for the potential artificial inflation of statistical power, we included a random effect on the interaction between network identity and insect species. This structure explicitly treats the specialization metrics as repeated measures taken on the same statistical unit.

In summary, while we acknowledge that this modeling approach is somewhat original, we believe it offers two key advantages: (i) presenting seven separate models /results would be confusing for the reader, and (ii) our approach remains statistically valid, as the issue of inflated power is adequately addressed within a repeated measures framework (Faraway, 2016).

In addition, as you noted in your comment, readers interested in metric-specific results can refer to Appendices 5 and 10 for the detailed analyses.

These justifications were added to the Methods section under network analysis.

Reference cited:

Faraway JJ. Extending the linear model with R : generalized linear, mixed effects and non parametric regression models. 2nd edition. Chapman & Hall/CRC Taylor & Francis group; 2016.

---

## [Decision Letter · Decision Letter 2]

29 Aug 2025

Insect-habitat-plant interaction networks provide guidelines to mitigate the risk of transmission of *Xylella fastidiosa* to grapevine in Southern France

PONE-D-24-38713R2

Dear Dr. Mesmin,

We’re pleased to inform you that your manuscript has been judged scientifically suitable for publication and will be formally accepted for publication once it meets all outstanding technical requirements.

Kind regards,

Tzen-Yuh Chiang

Academic Editor

PLOS ONE

Additional Editor Comments (optional):

Reviewer #2:

Reviewer #3:

Reviewers' comments:

Reviewer's Responses to Questions

**Comments to the Author**

Reviewer #2: All comments have been addressed

Reviewer #3: (No Response)

2. Is the manuscript technically sound, and do the data support the conclusions?

Reviewer #2: Yes

Reviewer #3: Yes

3. Has the statistical analysis been performed appropriately and rigorously?

Reviewer #2: Yes

Reviewer #3: Yes

4. Have the authors made all data underlying the findings in their manuscript fully available?

Reviewer #2: Yes

Reviewer #3: Yes

5. Is the manuscript presented in an intelligible fashion and written in standard English?

Reviewer #2: Yes

Reviewer #3: Yes

Reviewer #2: The authors addressed all the previous comments, providing further improvements to the manuscript. I commend the authors for their work.

Here are just a few inaccuracies I found in the article which should be fixed before its publication.

Lines 61-62. I think the correct subject of this sentence is Xf subsp. morus and not Xf subsp. sandyi. Moreover, data obtained by Vanhove et al. (2019, see https://doi.org/10.1128/AEM.02972-18) do not support that Xf subsp. morus is the result of a wide recombination between Xf subsp. fastidiosa and multiplex. Please correct the sentence and add appropriate reference.

Lines 68-69. I suggest to say that “The insect vectors of Xf are mainly xylem feeders…” because one known vector species, Cyphonia clavigera (Membracidae) is a phloem feeder [see https://gd.eppo.int/taxon/XYLEFA/vectors].

Reviewer #3: Comments and Suggestions for Authors

This work provides useful information about Xylella fastidiosa vectors habitat and hosts with a general view of the potential vector species and with a great variety of samplings points in three important winegrowing regions of France. Pearce’s disease is very severe and is a great threat for a key crop in Europe. The focus on vineyards is particularly interesting for the lack of studies about XF in vineyards in Europe because XF fastidiosa (Pierce’s disease causal agent) was restricted to the island of Mallorca till the detection of st1 XF in Southern Italy and Portugal the last years. The methodological work and data treatment is accurate and useful for the hypothesis evaluation, bringing clarity to the results in a type of study that is difficult to homogenize due to the variety of sampling points, hosts, insects and several factors more. We missed a longer sampling period in the study, about three years, which would have allowed for mitigating annual climate variability in the study. It would also have been interesting to sample in the summer, to obtain data from a period of high interest for the epidemiology of XF, but we are aware of the workload of maintaining the studio for years. The manuscript writing is accurate with no errors. For all these reasons, we recommend the publication of this manuscript in the present form.

General comments

It would be interesting to include in the introduction and discussion the experiences on the island of Mallorca, where there are field experiences with XF fastidiosa St1 in vineyards for longer time in Europe. I suggest some publications of XF vectors and pierce’s disease in Mallorca:

Moralejo et al., 2019 Insights into the epidemiology of Pierce's disease in vineyards of Mallorca, Spain. https://doi.org/10.1111/ppa.13076

Olmo et al., 2021 Landscape Epidemiology of Xylella fastidiosa in the Balearic Islands. https://doi.org/10.3390/agronomy11030473

Llompart et al., 2025. Spread of Xylella fastidiosa in Majorca Island Agroecosystems: Interaction Between Insect Vectors, Woody Hosts and Bacterial Subspecies. https://doi.org/10.1111/ppa.14135

Minor point

Line 533. Substitute “lever” for level.

**Do you want your identity to be public for this peer review?** For information about this choice, including consent withdrawal, please see our Privacy Policy

Reviewer #2: No

Reviewer #3: No

---

## [Editor Report · Acceptance letter]

PONE-D-24-38713R2

PLOS ONE

Dear Dr. Mesmin,

I'm pleased to inform you that your manuscript has been deemed suitable for publication in PLOS ONE. Congratulations! Your manuscript is now being handed over to our production team.

Kind regards,

on behalf of

Dr. Tzen-Yuh Chiang

Academic Editor

PLOS ONE